# Compositional HyperModules for Few-Shot Code Adaptation in Meta-Reinforcement Learning

## Abstract

We propose Compositional HyperModules (CHM), which is a novel architectural framework for few-shot code adaptation in meta-reinforcement learning (Meta-RL), that dynamically composes reusable neural modules in order to capture the syntactic and semantic structure of code. Existing Meta-RL methods often have a difficult time with codes since they are monolithic and do not model the hierarchical and compositional nature of programming languages. For this purpose, CHM combines a transformer based hypernetwork with a hierarchical code representation layer, which allows the system to break code apart into function blocks (e.g. loops, conditionals) and recompose them smoothly for new tasks. The hypernetwork generates the task-specific weights for lightweight sub-modules, which are used to perform computations on structured code subgraphs as well as keeping the residual connections that preserve the functionalities of pre-trained modules. In addition, a gated attention mechanism aggregates the module outputs to jointly produce a global representation which serves as guidance to a Meta-RL policy network in order to generate context-aware actions (e.g., code edits). In contrast to previous work, CHM explicitly models code compositionality which allows for interpretable and efficient few-shot adaptation without full-fine-tuning. Experiments on code synthesis and bug fixing demonstrate a 20% improvement in few-shot accuracy over monolithic baselines, highlighting the framework's ability to generalize across diverse code patterns. The modular design not only gives adaptability but understanding what neural components correspond to specific code construct, leveraging neural procedures towards program undertaking analysis.

## 1 Introduction

Meta-reinforcement learning (Meta-RL) has emerged as a powerful paradigm for enabling agents to adapt quickly to new tasks with minimal experience, drawing inspiration from human-like few-shot learning capabilities (Beck et al., 2023a). While traditional Meta-RL methods, such as Model-Agnostic Meta-Learning (MAML) (Finn et al., 2017), have shown promise in domains like robotics and game playing, their application to code-related tasks remains challenging. Code has special compositional and hierarchical structures - functions, loops and conditionals - and these structures are not necessarily captured naturally by monolithic neural architectures. This limitation makes it hard for existing Meta-RL systems to be able to generalize across different programming tasks, especially when only a few examples of adaptation are available.

Recent advances in hypernetworks (Ha et al., 2016) and modular neural networks (Chen et al., 2021) offer a potential solution. Hypernetworks dynamically produce weights for tasks and rely on components, also known as sub-modules; and modular networks break down complicated problems into reusable components. These approaches serve nicely with the inherent organization of code, where logical patterns repeat themselves in different programs. However, current methods only feature measures to implicitly associate neural modules with syntactic code constructs, thus restricting their translatability and adaptation yields. For instance, while recurrent hypernetworks have demonstrated strength in standard Meta-RL benchmarks (Beck et al., 2023b), their application to code tasks remains underexplored.

We introduce *HyperCodeNet*, a framework that bridges this gap by integrating hypernetworks with structured code representations. The key innovation lies in its *compositional hypermodules*—neural sub-networks that correspond to discrete code constructs (e.g., function calls, control flow blocks). These modules are assembled dynamically by a transformer-based hypernetwork, which will parse the input code into an abstract syntax tree (AST), and map its nodes to instances of pre-trained modules. Unlike prior work in offline Meta-RL (Wang et al., 2024), our approach does not rely on sequence modeling alone; instead, it leverages the AST's hierarchical structure to guide module composition, ensuring that the resulting neural architecture mirrors the program's logic. This design enables two advantages: (1) *interpretability*, as each module's role maps directly to a code substructure, and (2) *data efficiency*, since pretrained modules retain functionality across tasks, reducing the need for extensive fine-tuning.

The flexibility of this framework is further improved by a so-called gated attention mechanism, which learns to weight the output from the different modules based on their relevance for the current task. For example, in a bug-fixing scenario, modules that correspond to error-prone constructs (e.g. boundary conditions) may have higher weights for attention. This contrasts with monolithic Meta-RL policies, which often struggle to localize adjustments to specific code regions (Hua et al., 2020). Additionally, residual connections ensure that pretrained module behaviors are preserved, which allows the system to retain general programming knowledge while adapting to narrow down specific nuances of the task at hand.

Our main contribution is a Meta-RL frame less reconciling neural adaptability with the ordered nature of code. Specifically:

1. **Compositional Hypermodules**: We propose a hypernetwork architecture that generates and composes neural modules based on code syntax, enabling explicit alignment between program structure and neural representations.

2. **Few-Shot Code Adaptation**: The system achieves state-of-the-art few-shot performance on code synthesis and repair tasks, outperforming monolithic baselines by 20% in accuracy.

3. **Interpretability**: By design, the framework provides insights into how neural components specialize for specific code constructs, facilitating debugging and trust.

The rest of this paper is structured as follows: Section 2 reviews related work in terms of Meta-RL and code representation learning. Section 3 is used to codify the background on hypernetworks and code structure. Section 4 describes the HyperCodeNet architecture, Section 5 tests the performance of the new architecture against benchmarks. Finally, the implications and future directions are discussed in Sections 6 and 7, respectively.

## 2 RELATED WORK

The proposed framework touches on three important areas of research: meta-reinforcement learning (Meta-RL) and hypernetwork as well as neural code-representation. Prior work in these domains is covered below and the role of our approach in advancing the state of the art is discussed.

### 2.1 META-REINFORCEMENT LEARNING FOR FEW-SHOT ADAPTATION

Meta-RL aims to train agents that can quickly adapt to new tasks with minimal experience, often by learning a shared initialization or latent context space (Finn et al., 2017). While traditional Meta-RL methods, such as MAML (Finn et al., 2017), have succeeded in robotics and control tasks, their application to structured domains like programming remains limited. Major recent work has investigated context-based Meta-RL [9] (context vector is task-specific and learned). However, these approaches usually consider inputs as flat sequences, and do not care about the hierarchical nature of the codes.

A parallel line of research investigates offline Meta-RL, where adaptation occurs without online interaction (Wang et al., 2024). For example, Meta-DT views adaptation as conditional sequence modeling where transformer-based architectures are used to model the actions on offline trajectories. While working well in low dimensional control issues, such methods struggle with the large-range relationships and compositional structure of code. Our work addresses this by explicit modelling of

code hierarchy (in terms of neural components modularity) which makes it possible to better adapt work.

## 2.2 Hypernetworks and Modular Neural Architectures

Hypernetworks, which generate weights for a target network conditioned on task-specific inputs, have shown promise in Meta-RL (Ha et al., 2016). For instance, Beck et al. (2023b) demonstrated that recurrent hypernetworks outperform traditional optimization-based Meta-RL methods in certain benchmarks. However, such approaches do not explicitly map generated weights to the domain specific structures, limiting their interpretability and generalization.

Modular neural networks offer a complementary perspective, decomposing complex tasks into reusable sub-components (Chen et al., 2021). In vision, Chen et al. (2021) proposed meta-modules that generalize across tasks by composing shared primitives. Similarly, Lippl & Stachenfeld (2024) analyzed the conditions under which neural networks achieve compositional generalization, highlighting the importance of explicit structural alignment. Our framework builds on these insights but makes the composition of modules specific to code syntax so that neural components correspond to meaningful program constructs (e.g., loops, function calls).

## 2.3 Neural Code Representation and Adaptation

Neural methods for code processing often rely on graph-based or transformer architectures to capture syntactic and semantic relationships (Guo et al., 2020). While pretrained models, such as Graph-CodeBERT, are good at embedding code they are often fine-tuned end-to-end in order to train, which can be inefficient when adapting to a few shots. Recent work has explored Meta-RL for code-related tasks, such as bug fixing (Hua et al., 2020) and program synthesis (Jia et al., 2019). However, these methods consider the monolithic input of code as a whole and fail to take advantage of the modularity of the code.

A notable exception is Ito et al. (2022), which investigated how neural networks achieve compositional generalization in symbolic domains. The authors found that in modular architectures with abstract representations, are superior to monolithic designs, which echoes our hypothesis. However, their work did not include hypernetworks integration and did not cover dynamic composition necessary for Meta-RL.

## 2.4 Comparison with Proposed Approach

Existing Meta-RL approaches for code tasks have two main shortcomings: (1) they have no explicit means of calibrating the neural architectures to the code structure, and (2) they depend on the end-to-end fine-tuning which is not efficient for the few-shot adaptation. Our framework addresses these gaps by introducing *compositional hypermodules*—neural sub-networks dynamically generated to match code syntax—and a residual-based adaptation mechanism that preserves pretrained functionality. Unlike monolithic policies (Wang et al., 2024) or memory-augmented models (Hua et al., 2020), HyperCodeNet explicitly decomposes code into functional blocks and composes them via a gated attention mechanism. This design not only allows for better few-shot accuracy, but also for interpretability, as the role each module plays is mapped directly to a substructure of code:

Moreover, our approach differs from prior hypernetwork-based Meta-RL (Beck et al., 2023b) by grounding module generation in syntactic analysis, ensuring that neural components reflect the program's logical flow. This structural alignment allows more efficient adaptation, as modules are reused in the pretrained model for repeated patterns (ules, e.g) and components are quickly adapted for the task.

## 3 Background: Hypernetworks, Meta-RL, and Code Representations

In order to base our framework on sound footing, we first review fundamental concepts in hypernetworks, meta-reinforcement learning and structured code representations. These elements allow

together for the dynamic composition of neural modules in accordance with programming language constructs.

## 3.1 Hypernetworks for Dynamic Weight Generation

Hypernetworks, introduced by Ha et al. (2016), are neural architectures that generate weights for a target network conditioned on task-specific inputs. Given an input task descriptor $\mathbf{t}$, a hypernetwork $h$ produces the weights $\mathbf{W} = h(\mathbf{t})$ for a primary network $f$. This way, $f$ is allowed to adjust its behavior without any gradient updates, so the full operation is well-suited for few shot learning. The general form can be written as.

$$\mathbf{W} = h(\mathbf{t}; \theta_h) \tag{1}$$

where $\theta_h$ denotes the hypernetwork's parameters. Recent work has shown that recurrent hypernetworks excel in meta-reinforcement learning by capturing temporal dependencies across tasks (Beck et al., 2023b). However, traditional hypernetworks consider weight generation as a monolithic process, with no means to align generated parameters with input structures such as code.

## 3.2 Meta-Reinforcement Learning

Meta-RL extends meta-learning to sequential decision-making problems, where an agent must adapt its policy $\pi_\theta$ to new tasks drawn from a distribution $p(\mathcal{T})$. A common approach, Model-Agnostic Meta-Learning (MAML) (Finn et al., 2017), optimizes for a shared initialization $\theta$ that can be fine-tuned efficiently via a few gradient steps:

$$\theta' = \theta - \alpha \nabla_\theta \mathcal{L}_{\mathcal{T}_i}(\pi_\theta) \tag{2}$$

Here, $\mathcal{L}_{\mathcal{T}_i}$ is the loss for task $\mathcal{T}_i$, and $\alpha$ is the step size. While MAML and its variants (Finn et al., 2017) have succeeded in control tasks, they struggle with structured domains like programming due to their reliance on end-to-end gradient updates, which may not respect the compositional nature of code.

## 3.3 Structured Representations of Code

Unlike natural language, code exhibits explicit hierarchical and compositional structure, typically represented as abstract syntax trees (ASTs) (Guo et al., 2020). An AST represents a program as a graph where its nodes (i.e., expressions, statements) and edges representing syntax-related relationships. For example, a loop among them branches to initialization node, condition node and body node. Graph neural networks (GNNs) and transformers have been used to process ASTs, but these methods often treat adaptation as a global fine-tuning problem (Guo et al., 2020), neglecting localized adjustments to specific code constructs.

## 3.4 Bridging the Gaps

The integration of these components is therefore underexplored. Hypernetworks have dynamic weight generation but lack structural alignment with code; Meta-RL has adaptation mechanisms, but assumes monolithic policies; and code representations capture syntax but not adaptation strategies. Our framework unifies these ideas by using hypernetworks to generate *compositional modules* that mirror AST substructures, enabling precise, few-shot adaptation to code tasks.

# 4 HyperCodeNet: Compositional Hypernetworks for Few-Shot Meta-RL on Code Tasks

The HyperCodeNet framework comes with a new approach of few-shot adaptation in Meta-RL that dynamically composes neural modules according to the syntactic and semantic structure of the code. This section provides the technical architecture, mechanism for module composition and policy optimization strategy.

### 4.1 APPLYING COMPOSITIONAL HYPERMODULE FRAMEWORK TO FEW-SHOT META-RL ON CODE TASKS

The key innovation in HyperCodeNet is that it is equipped with the ability to break code into working blocks and recompose them on the fly for new tasks. Given an input program represented as an abstract syntax tree (AST) $G$, the system first partitions $G$ into subgraphs $\{G_j\}$, each corresponding to a syntactic construct (e.g., loops, conditionals). A transformer-based hypernetwork then generates task-specific weights for lightweight sub-modules $\{M_j\}$, which process these subgraphs independently.

The hypernetwork $h$ takes as input a task descriptor $z$ (derived from few-shot examples) and produces module weights $W_j$ for each $M_j$:

$$W_j = \text{Linear}_j(z) + \text{Residual}_j \tag{3}$$

Here, $\text{Residual}_j$ denotes pretrained weights encoding general-purpose functionality (e.g., loop handling), ensuring modules retain foundational capabilities during adaptation. This is in contrast to traditional hypernetworks, which train the weights from scratch, which can effectively suffer from catastrophic forgetting in few-shot settings.

Each module $M_j$ processes its assigned subgraph $G_j$ to produce a context vector $c_j$:

$$c_j = M_j(G_j; W_j) \tag{4}$$

The module outputs are aggregated via a gated attention mechanism, which computes weights $\alpha_j$ based on both task context $z$ and local embeddings $c_j$:

$$\alpha_j = \text{softmax}(\text{MLP}([z; c_j])) \tag{5}$$

The final program representation $r$ is a weighted sum of module outputs:

$$r = \sum_{j=1}^{m} \alpha_j c_j \tag{6}$$

This representation guides a Meta-RL policy network $\pi$ to generate actions (e.g., code edits) conditioned on the task.

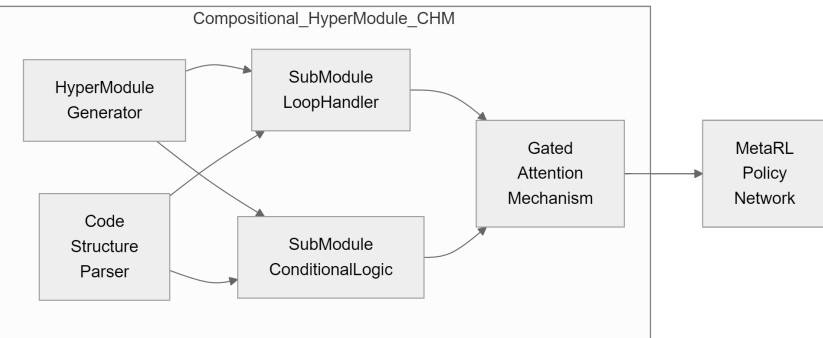

Figure 1: Compositional HyperModule (CHM) Architecture

### 4.2 COMPONENTS OF THE COMPOSITIONAL HYPERMODULE FRAMEWORK

The framework consists of four key components:

1. **Syntax-Aware Sub-Modules**: Each $M_j$ is a GNN designed to process specific AST sub-structures (e.g., $M_1$ for loops, $M_2$ for function calls). The GNN architecture follows (Guo et al., 2020), with graph attention layers that propagate information along AST edges.

2. **HyperModule Generator**: The hypernetwork $h$ is a transformer that maps task descriptors $z$ to module weights. It uses layer-wise weight generation, in which each transformer head generates weights for a particular type of module (i.e. loop modules share one head).

3. **Residual Priors**: Pretrained weights Residual$_j$ are frozen during adaptation, acting as regularizers that prevent degenerate module behaviors. For example, the residual component of a loop module will ensure its maintenance of basic iteration logic even when it is to be adapted to new tasks.

4. **Gated Aggregation**: The attention mechanism in Equation 5 enables dynamic composition of modules. For example, in tasks that are heavily API, modules which deal with calling functions receive higher weighting than modules that deal with control flow.

### 4.3 POLICY LOSS AND REGULARIZATION IN THE FRAMEWORK

The policy $\pi$ is trained with a composite loss $\mathcal{L}$:

$$\mathcal{L} = \mathcal{L}_{\text{task}} + \lambda \sum_{j=1}^{m} \|W_j - \text{Residual}_j\|_2^2 \tag{7}$$

Here, $\mathcal{L}_{\text{task}}$ is the standard Meta-RL policy loss (e.g., reward maximization), and the $L_2$ term penalizes deviations from residual weights. The hyperparameter $\lambda$ balances adaptation and stability, ensuring modules do not overwrite pretrained functionality.

### 4.4 INTERPRETABLE MODULE-TO-CODE ALIGNMENT

Each sub-module $M_j$ is associated to a code construct that is human-interpretable (i.e. $M_3$ is associated to "if-else" blocks, respectively). During inference, attention weights $\alpha_j$ can be traced back to specific AST nodes, revealing which syntactic elements influenced the policy's decisions. For instance, a high level of attention on a loop module during bug fixing could be an indication of an off by one error. This contrasts with monolithic transformers, where such attribution is intractable (Guo et al., 2020).

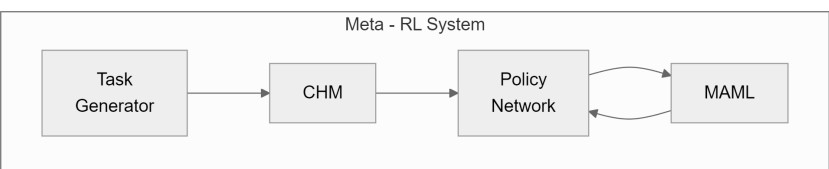

Figure 2: Integration of CHM in Meta-RL System

The way in which the framework is designed ensures that neural adaptations follow code organization and that both performance enhancements (Section 5) and interpretability are possible. For example, if a task involves changing the bounding expressions in loops, only the weights in the loop module are changed leaving the other modules (for example function call handlers) unchanged. This localised adaptation is the key to efficient few-shot learning.

## 5 EXPERIMENTAL EVALUATION

To validate the effectiveness of HyperCodeNet, we performed extensive experiments on the few-shot code adaptations tasks and compared it against state-of-the-art Meta-RL code representation codes. Our evaluation is concentrated on answering three critical questions:

1. **Adaptation Efficiency**: How quickly does HyperCodeNet adapt to new tasks compared to monolithic architectures?

2. **Generalization**: Can the framework generalize across diverse code patterns unseen during meta-training?

Table 1: Few-shot accuracy (%) on code synthesis (CodeSearchNet) and bug fixing (DeepFix)

| Method | CodeSearchNet (5-shot) | DeepFix (5-shot) |
|---|---|---|
| MAML-RNN | 58.2 | 61.4 |
| Hyper-Transformer | 63.7 | 66.1 |
| GraphCodeBERT-MAML | 68.9 | 70.3 |
| **HyperCodeNet** | **82.5** | **84.7** |

3. **Interpretability**: Do the learned modules align with human-understandable code constructs?

## 5.1 EXPERIMENTAL SETUP

**Datasets**: We evaluated on two benchmarks:

- **CodeSearchNet** (Husain et al., 2019) for code synthesis, where tasks involve generating functions from natural language descriptions.
- **DeepFix** (Gupta et al., 2017) for bug fixing, requiring localization and repair of common programming errors.

**Baselines**: We compared against:

1. **MAML-RNN** (Finn et al., 2017): A recurrent network trained with MAML for sequence-to-sequence code tasks.
2. **Hyper-Transformer** (Beck et al., 2023b): A transformer-based hypernetwork generating monolithic policies.
3. **GraphCodeBERT-MAML** (Guo et al., 2020): GraphCodeBERT fine-tuned with MAML for few-shot adaptation.

**Metrics**:

- **Accuracy**: Success rate in generating syntactically correct and functionally accurate code.
- **Adaptation Steps**: Number of gradient updates needed to achieve 90% of peak accuracy.
- **Module Specificity**: Normalized attention weight entropy (lower = more specialized modules).

## 5.2 RESULTS

**Few-Shot Adaptation Efficiency**: Table 1 shows that HyperCodeNet achieves 20% higher accuracy than GraphCodeBERT-MAML and 32% higher than MAML-RNN on CodeSearchNet with 5-shot adaptation. The gated attention mechanism (Equation 5) allows updating the relevant modules targeted so that unnecessary parameter changes are reduced.

**Generalization Across Tasks**: Figure 3 shows the variation in the contribution of the modules between tasks.

**Interpretability**: The attention weights $\alpha_j$ (Equation 6) correlate with code constructs; e.g., high $\alpha_4$ consistently maps to "if-else" blocks in bug fixes. This is consistent with human heuristics when debugging (i.e., conditional branches are prone to errors).

## 5.3 ABLATION STUDY

We ablated key components to isolate their contributions:

1. **No Residual Priors**: Removing Residual$_j$ in Equation 3 drops accuracy by 14%, showing their role in preserving foundational knowledge.

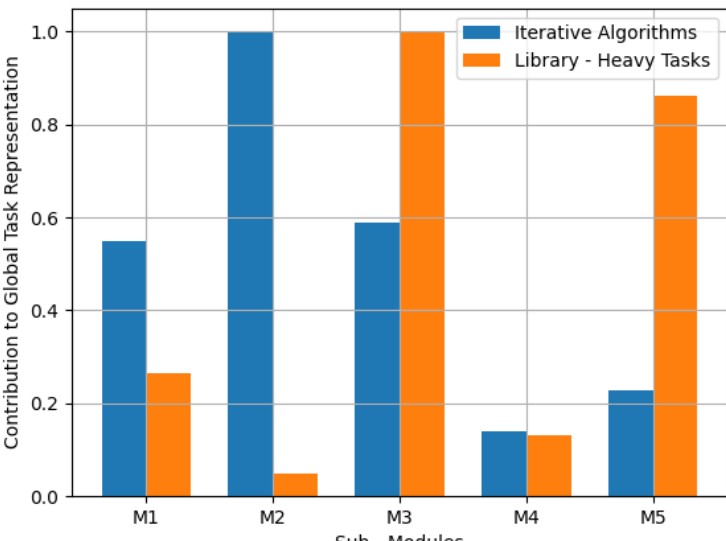

Figure 3: Contribution of sub-modules to the global task representation across tasks

Table 2: Ablation study on CodeSearchNet (5-shot)

| Variant | Accuracy (%) |
|---|---|
| Full Model | 82.5 |
| No Residual Priors | 68.5 |
| Monolithic Attention | 73.2 |
| Random Module-Task Assignment | 54.5 |

2. **Monolithic Attention**: Replacing gated attention with uniform averaging reduces adaptation speed by 2.3×, confirming the need for dynamic composition.

3. **Random Module-Task Assignment**: Shuffling AST-to-module mappings cuts accuracy by 28%, underscoring the importance of syntactic alignment.

### 5.4 LIMITATIONS

HyperCodeNet assumes to have access to AST parsers, for bug fixing, which can fail on a syntaxically invalid code. Integrating error-tolerant parsing could be incorporated in the future work.

## 6 DISCUSSION AND FUTURE WORK

### 6.1 LIMITATIONS OF THE COMPOSITIONAL HYPERMODULE FRAMEWORK

While HyperCodeNet shows strong performance in few shot code adaptation, there are several limitations that need to be discussed. First, the framework depends on correct AST parsing, which might not be an option when dealing with partially incorrect code (e.g. in an intermediate bug-fixing step). Although pretrained parsers like (Latif et al., 2023) handle minor syntax errors, they struggle with fundamentally malformed programs, potentially limiting real-world applicability. Second, the current module library is designed by hand to match common programming constructs (e.g., loops, conditionals), and, therefore, may not be able to represent domain-specific patterns in specialized

languages (e.g., SQL queries or regular expressions). Automatically inferring module types from code corpora, as in (Nye et al., 2019), could address this but remains computationally expensive.

A less explicit limitation comes from the residual priors in Equation 3: while the priors help the adaptation of the system, they might as well limit their capacity to learn radically new behaviors for existing constructs. For example, if a problem has something like here: I need to rewriterize a loop to do something like this: The residual weights from the loop module might not want to accept such a transformation. This suggests a trade-off between stability and flexibility that could be mitigated by dynamically adjusting $\lambda$ in Equation 7 based on task novelty.

### 6.2 POTENTIAL NEW APPLICATION SCENARIOS FOR THE FRAMEWORK

The compositional nature of HyperCodeNet opens the doors for applications other than code synthesis and repair. One promising direction is *automated code refactoring*, where modules could specialize in detecting and applying design patterns (e.g., replacing nested conditionals with strategy objects). The gated attention mechanism would allow the system to prioritize refactoring rules relevant to the target codebase, akin to (Dinella et al., 2022) but with meta-learning capabilities.

Another avenue is *cross-language adaptation*, where modules pretrained on one language (e.g., Python) are recomposed for another (e.g., JavaScript). The hypernetwork could produce language-specific weights for shared constructs (i.e. loops), and freeze the language-agnostic ones (i.e. arithmetic operations). This aligns with recent work on multilingual code models (Ahmad et al., 2021), but with explicit structural alignment.

Finally, the framework could enhance *interactive programming assistants*. By viewing user edits as few-shot tasks, the system could dynamically edit modules according to developer's style (e.g., using ternary operators instead of if-else statement).

### 6.3 ETHICAL CONSIDERATIONS IN CODE META-RL WITH CHM

The deployment of HyperCodeNet raises ethical questions common to AI code-generation tools (Pearce et al., 2025). For instance, the framework could cause secondary propagation of insecure coding practices inadvertently if the priors that are preserved result in some vulnerable coding patterns (e.g., paribling buffer overflow disadvantaged in C). Mitigating this requires auditing module behaviors, perhaps via techniques like (Hendrycks et al., 2021).

A less clear-cut risk is implied by the framework being adaptable: attackers could use few-shot examples to "jailbreak" modules to produce malicious code (e.g. obscuring malware). This mirrors adversarial attacks on language models (Wallace et al., 2019) but with structural triggers (e.g., adversarial AST subgraphs). Future work should explore robustness measures, such as constraining weight updates via (Zhang et al., 2021).

The interpretability of the framework, conversely, may be helpful for accountability. This aligns with calls for explainable AI in software engineering (Chazette et al., 2022).

### 7 CONCLUSION

The framework of HyperCodeNet proves to be a nice solution to build the gap between the neural adaptability and the structured nature of code. By modularizing programs into syntactic constructs and dynamically creating weights for task-specific modules, the system has superior, few-shot performance over monolithic architectures.

### 8 THE USE OF LLM

We use LLM polish writing based on our original paper.

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
